# Implication of *Urochloa* spp. Intercropping and Conservation Agriculture on Soil Microbiological Quality and Yield of Tahiti Acid Lime in Long Term Orchard Experiment

**Ana Carolina Costa Arantes** [1,*], **Simone Raposo Cotta** [2], **Patrícia Marluci da Conceição** [3], **Silvana Perissatto Meneghin** [4], **Rodrigo Martinelli** [1], **Alexandre Gonçalves Próspero** [1], **Rodrigo Marcelli Boaretto** [1], **Fernando Dini Andreote** [5], **Dirceu Mattos-Jr.** [1] **and Fernando Alves de Azevedo** [1]

[1] Instituto Agronômico, Centro de Citricultura Sylvio Moreira, Anhaguera Road, Km 158, PoBox 04, Cordeirópolis, SP 13490-970, Brazil; rodrigo_martinelli@hotmail.com (R.M.); alexandregprospero@gmail.com (A.G.P.); boaretto@ccsm.br (R.M.B.); ddm@ccsm.br (D.M.-J.); fernando@ccsm.br (F.A.d.A.)

[2] Centro de Energia Nuclear na Agricultura, Universidade de São Paulo, Centenário Avenue, n°303, Piracicaba, SP 13400-970, Brazil; raposo.cotta@gmail.com

[3] Departamento de Desenvolvimento Rural, Centro de Ciências Agrárias, Universidade Federal de São Carlos, Anhanguera Road, Km 174, Araras, SP 13600-970, Brazil; patymarluci@gmail.com

[4] Departamento de Biotecnologia Produção Vegetal e Animal, Centro de Ciências Agrárias, Universidade Federal de São Carlos, Anhanguera Road, Km 174, Araras, SP 13600-970, Brazil; silvanapmeneghin@gmail.com

[5] Departamento de Ciência do Solo, Escola Superior Luiz de Queiroz, Universidade de São Paulo, Pádua Dias Avenue, n°11, Piracicaba, SP 13418-260, Brazil; fdandreo@gmail.com

* Correspondence: accarantes@gmail.com

**Abstract:** Techniques such as intercropping and minimum tillage improve soil quality, including soil microbial activity, which stimulates the efficient use of soil resources by plants. However, the effects of such practices in soil under citrus orchards have not been well characterized. In this study, we aimed to determine the effects of mowing and intercrop species on soil microbiological characteristics beneath a Tahiti acid lime orchard. The orchard was planted using minimum tillage and intercropped with two species of *Urochloa* species (*U. ruziziensis*—ruzi grass; *U. decumbens*—signal grass), with two types of mowers for *Urochloa* biomass (ecological; conventional) and herbicide applications. The study was conducted over 10 years. The ecological mower made the largest deposition of the intercrop biomass, thus providing the lowest disturbance of soil microbial activity and increasing, on average over all 10 years, the basal soil respiration (45%), microbial biomass carbon (25%), abundance of 16S rRNA (1.5%) and ITS (3.5%) genes, and arbuscular mycorrhizal fungi (30%), and providing a ca. 20% higher fruit yield. *U. ruziziensis* in combination with ecological mowing stimulated the abundance of the genes *nif*H (1.5%) and *pho*D (3.0%). The herbicide showed little influence. We conclude that the use of *U. ruziziensis* as an intercrop in citrus orchards subjected to ecological mowing can be recommended for improving and sustaining soil quality and citrus fruit production.

**Keywords:** mulching; cover crops; conservation agriculture; citrus; soil microbial community

## 1. Introduction

The quality of soil is related to the capacity to sustain crop yields and to provide health for plants, animals, and humans [1]. In agriculture, this quality is associated with the capacity of the soil to supply nutrients to plants, thus facilitating their growth and development [2].

Methods for assessing soil quality are related to cultural practices, which promote soil health or functionality through physical, chemical, and microbiological methods [3]. The abundance and diversity of groups of species and communities, either absent or present, may indicate the state of the soil [4]. This evaluation can be carried out using genetic microbial material [5], such as the microorganisms responsible for the conversion of atmospheric nitrogen ($N_2$) into inorganic forms assimilated by the plants ($NO_3^-/NH_4^+$) [6].

Soil microbial biomass is the labile fraction of soil organic matter, accounting for 2% to 5% of the labile carbon in tropical soils [7]. The functions of microorganisms in soil are diverse, such as providing and maintaining the flow of energy and biochemical processes, acting on the dynamics of organic matter [8] by decomposing and mineralizing residues, and regulating nutrient flow [9]. These functions are able to provide stability and soil conservation, culminating in increased yields of agricultural crops [10]. Microbiological activity and microbial community composition are dependent on surface residues, which can alter enzymatic activities via carbon and nitrogen contents. In areas with a record of cultivating plants such as crotalaria, there was an increase in the community of diazotrophs [11].

The maintenance of plant residues in soil can cause minor disturbances in the anthropic environment, with less interference the soil microbial communities through a more balanced and sustainable environment [11]. Changes in natural or managed environments are noticeable in soil microbial variables, which respond quickly to different soil uses [12] with changes in their community structures.

The use of the herbicide glyphosate in citrus rows is a common practice in citrus orchards [13,14]. However, despite its biodegradation in soil, its effect on the shikimic acid pathway can cause disturbances in the cellular metabolism of bacteria, fungi, plants, and other organisms [15], and can harm the composition of the soil microbial community [16].

In tropical environments, such as the southeast of Brazil, the incorporation of vegetable residues into soil is dependent on the quantity and quality of the cultivated plant species, since the decomposition process of organic materials is accelerated, which can increase carbon losses [17]. In systems that aim at maintaining and injecting organic material into the soil [3] and at increasing the organic carbon content both on and below the surface, it is recommended to plant species of the genus *Urochloa*, for example in citrus orchards [18].

Conservation agriculture techniques, which aim at minimum soil tillage, permanent soil cover, and species diversification with intercrops, can favor microbiological soil activity [19]. These techniques should be used to form a sustainable environment, with interactions of beneficial microorganisms and stable crop yield. Such an approach will help to meet the requirements of the consumer market, which is increasingly looking for quality food from sustainable environments [10].

Brazil, after China, is the second largest citrus producer in the world, producing 19.6 million tons per year, and the fifth largest producer of lemons and limes, at 1.3 million tons per year [20]. However, the lack of sustainable practices by the vast majority of lemon and lime producers [21], who usually use harrows and plows and do not cover the soil in the inter-rows [13], accelerates the decomposition of organic matter and the loss of microbial biomass [8]. Furthermore, intense machine traffic in orchards has resulted in soil loss and reduced microbiological quality, with a consequent limitation in fruit yield.

In citrus orchards, conservation agriculture can be applied via intercropping with Poaceae spp. such as *U. decumbens* and *U. ruziziensis*. These are sown at the beginning of the planting in the citrus inter-rows and are managed throughout the cycle of the main crop; after cutting, the biomass is deposited in the intra-rows, by the lateral "ecological" type of mower [14]. Benefits of this system were found in the cultivation of Pera sweet orange, in which green manure of the winter biomass deposited in the intra-row resulted in increased yields [22]. Soil microbiological indicators can be used to assess

soil quality, as they can assist in increasing the yield of the Pera orange due to the improvement of the soil's microbiological activity [23].

Changes in the soil microbial community should be studied over time to understand the cover crop residues. In this context, the aim of this work was to compare the long-term effects of different treatments in *Urochloa* intercropping with Tahiti acid lime plants on soil microbiological attributes and Tahiti acid lime fruit yield.

## 2. Materials and Methods

### 2.1. Experimental Conditions

The experiment was installed in Mogi Mirim in the state of São Paulo, Brazil (22°24′ S 47°05′ W). The climate is of the Cwa type (subtropical with dry winters and hot and humid summers), according to the scale of Köppen–Geiger [24].

The soil was classified as Red-Yellow Argisol [25] or Cutanic Acrisol [26], and the chemical characteristics prior to planting in the layer of 0.0–0.2 m were as follows: SOM (soil organic matter; 23.0 g $dm^{-3}$), pH (potential hydrogen; 5.7 $CaCl_2$), P (phosphorus; 41.0 mg $dm^{-3}$), K (potassium; 1.3 $mmol_c$ $dm^{-3}$), Ca (calcium; 42.0 $mmol_c$ $dm^{-3}$), Mg (magnesium; 11 $mmol_c$ $dm^{-3}$), B (boron; 0.6 mg $dm^{-3}$), Zn (zinc; 3.5 mg $dm^{-3}$), H + Al (hydrogen + aluminum; 12.0 $mmol_c$ $dm^{-3}$), BS (base saturation; 54.3 $mmol_c$ $dm^{-3}$), CEC (cation exchange capacity; 66.3 $mmol_c$ $dm^{-3}$), and V (base saturation; 82%).

In 2009, two species of tropical Poaceae (grasses), *U. ruziziensis* (R. German and Evrard) and *U. decumbens* (Stapf), in densities of 12 and 14 kg $ha^{-1}$, respectively, were sown. In 2010, plants of the Tahiti acid lime—IAC 05 (*Citrus latifolia* (Yu. Tanaka) Tanaka), grafted onto *Swingle citrumelo* (*C. paradisi Macf.* × *Poncirus trifoliata* (L.) Raf.), were planted using minimum tillage in a randomized complete block in a split-split-plot design (2 × 2 × 2), with three replications. The spacing was 7.0 m (inter-row) by 4.0 m (intra-row, between plants), and each plot was composed of 24 trees, distributed in three rows with eight trees each and three replications, with a total of 72 trees per treatment. The evaluations were concentrated on the four central trees of the central row.

The treatments were distributed as follows: the plot represented the two species of tropical Poaceae, *U. ruziziensis* (RUZ), and *U. decumbens* (DEC). In the subplots we represented the management of these *Urochloa* spp., either via a conventional mower (CONV), which cuts the biomass and allows the residue to remain in the inter-row (between citrus rows) area (Figure 1A), or with an ecological mower (ECO), which cuts in the inter-row area and launches the resulting biomass into the intra-row area [14] (Figure 1B).

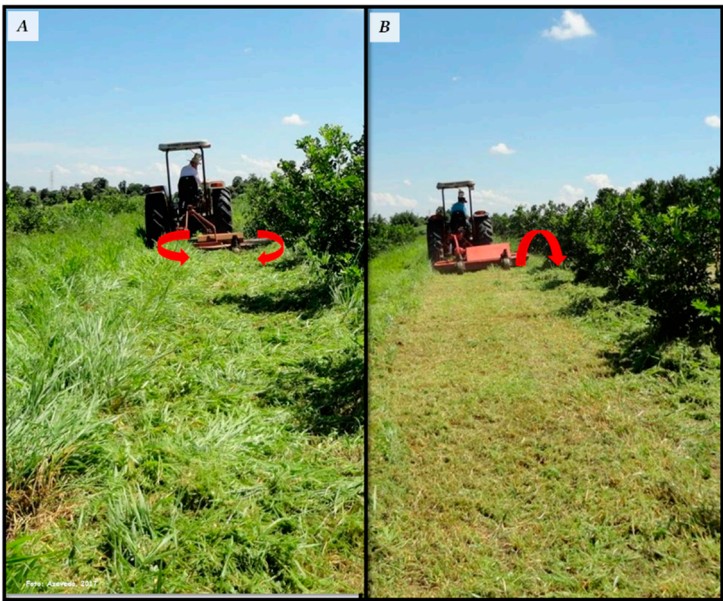

**Figure 1.** Conventional mower (CONV) (**A**) and Ecological mower (ECO) (**B**).

The sub-subplots were represented by either the use of glyphosate (H) or the absence of glyphosate (NH), with 3 L ha$^{-1}$; 1080 g ae ha$^{-1}$, on the intra-row. Mowing of *Urochloa*, with the ECO and the CONV, took place in the pre-flowering phase in summer, with the plants being in the maximum vegetative development stage. Glyphosate was applied in the planting line prior to mowing.

### 2.2. Evaluations

#### 2.2.1. Biomass of *Urochloa* Species

The biomass yield of the two species of *Urochloa* was quantified after biomass sampling in the inter-rows prior to mowing. Mowing (conventional and ecological) was carried out in the pre-flowering phase of the grass (three to four times a year), and afterwards the biomass deposited in the intra-row was assessed. For the two evaluations, a quadrant of 0.25 m$^2$ was established, with four repetitions in the inter-row (yield) and four repetitions in the intra-row (deposition), totaling 1.0 m$^2$ of sample area per plot. The samples were weighed and dried in a forced air oven at 60 °C ± 3 °C for 72 h. Subsequently, the yield and biomass deposition of grass (t ha$^{-1}$) were calculated for all 7 years (2012–2018).

#### 2.2.2. Soil Microbiological Analyses

Between February and March, six soil samples were collected and taken under the canopy projection of the Tahiti acid lime plants, from the surface layer (0.0–0.10 cm) for each treatment and were combined to form a composite sample. Evaluations of basal soil respiration (BSR) and microbial biomass carbon (MBC) took place in the fourth, sixth, seventh, and ninth years after planting. Abundance analyses and community structure of the 16S rRNA (bacterial) and ITS-Internal Transcribed Spacer (fungal) genes were carried out in the sixth, seventh, and ninth years after planting, and for abundance of the *nif*H (diazotrophs) and *pho*D (phosphate mineralizers) genes, the evaluations were carried out in the ninth year (2019).

The basal soil respiration (BSR) was evaluated by the "static system" [27], by capturing $CO_2$ from microbial soil respiration through extraction with 0.5 M NaOH (10.0 mL) and transformation to $NA_2CO_3$ [28], with incubation for 10 days [29] and was calculated using the formula:

$$BSR = \frac{(Vb - Va) \times M \times 22}{Dw}$$

**Theorem 1.** *Basal soil respiration (BSR), expressed in milligrams of $CO_2$ per g dry soil (mg $CO_2$ $g^{-1}$ solo); Va = volume of spent HCl in the sample titration; Vb = volume of spent HCl spent in the blank sample titration; M = exact molarity of HCl (0.5); Dw = dry weight of 50 g of sample wet soil, dried at 105 °C for 24 h.*

The microbial biomass carbon (MBC) was determined by the "fumigation-extraction method" [30], through the extraction of C by potassium dichromate ($K_2Cr_2O_7$) and $H_2SO_4$ + $H_3PO_4$ from fumigated (incubation with chloroform) and non-fumigated soil, and was calculated by:

$$MBC = \frac{\frac{Ta - Tb}{Kc}}{1000}$$

**Theorem 2.** *Microbial biomass carbon (MBC), expressed in milligrams of C per gram of dry soil (mg $g^{-1}$ of C); Ta = titration of fumigated samples with ferrous ammoniacal sulfate + sulfuric acid ($FeH_{20}N_2O_{14}S_2$ + $H_2SO_4$), mL; Tb = titration of non-fumigated samples, mL; Kc = correction factor, 0.33.*

The metabolic quotient ($qCO_2$), used to measure the efficiency of soil management, was calculated by the direct ratio of basal soil respiration per unit of microbial biomass carbon [2]:

$$qCO_2 = \frac{BSR}{MBC}$$

**Theorem 3.** *Metabolic quotient ($qCO_2$), expressed in milligrams of $CO_2$ per milligram C (mg $CO_2$ mg C $g^{-1}$ soil); BSR = basal soil respiration (mg $CO_2$ $g^{-1}$ soil); MBC = microbial biomass carbon (mg C $g^{-1}$ soil).*

To asses microbial community structure and abundance, the DNA was extracted from 0.4 g of each soil sample (total 64 samples), using the Powersoil™ DNA isolation Kit (MoBio/Qiagen Laboratories, Germantown, MD, USA), according to the manufacturer's instructions. The DNA preparations were visualized by electrophoresis in 1% agarose gel in 1 × TAE (Tris-Acetate-EDTA Buffer) to assess yield and integrity.

The DNA samples stored at −20 °C were used for analysis of the microbial community structure via terminal restriction fragment length polymorphism (T-RFLP) analysis, applying as target genes the bacterial 16S rRNA gene and fungal ITS regions. For bacteria, the 16S rRNA was amplified with primers 8fm and 926r. The forward primer, 8fm, was labelled with 6-FAM, and the PCR conditions are described in [31]. After amplification, the amplicons were fragmented using the endonuclease HhaI as described by the manufacturer (Thermo Fischer Scientific). For fungi, the ITS region was amplified with primers ITS1F and ITS4. The primer ITS1F was labeled with 5-FAM, and the PCR conditions are described in [32,33]. After amplification, the obtained products were digested with the endonuclease HaeIII as described by the manufacturer (Thermo Fischer Scientific).

The fragments generated were precipitated with ethanol/EDTA/sodium acetate, according to the protocols of the BigDye Terminator v. 3.1 Cycle Sequencing Kit (Thermo Fisher, Waltham, MA, USA). Analysis of the terminal restriction fragment (T-RF) size and quantity was performed on a 3500 Genetic Analyzer (Applied Biosystems, Foster City, CA, USA) and GS600LIZ (Life Technologies, Foster City, CA, USA) was used as a marker. The results were examined with GeneMapper® 4.1 software, with a baseline limit of 100 fluorescence units to remove the background noise, and the peak intensity tables were exported for further statistical analysis [34].

For T-RFLP analysis, the peak intensity (Figures S1 and S2) tables were exported and the patterns were analyzed, combined with the environmental parameters and redundancy analysis (RDA), conducted in CANOCO® [35].

Quantification for each sample was carried out twice using the StepOne Real Time System® (Applied Biosystems) with SYBR Green to determine the abundance of total bacteria and fungi,

and diazotrophs and phosphate mineralizers (Figure S3). Bacterial abundance was estimated by using the 16S rRNA partial gene as a proxy. The reaction was performed in 25.0 μL of reaction mixture containing Power SYBR Green PCR Master Mix (Applied Biosystems, Frankfurt, Germany), 0.5 μL of each primer (100.0 μm), and 1.0 μL of target DNA (≈10 ng). The primers used were P1/P2 [36], and the thermal cycling conditions were 35 cycles at 94 °C for 30 s for denaturation, followed by 55 °C for 30 s of annealing and 72 °C for 30 s as the final extension step. The standard curve was constructed using serial dilutions ($10^8$ to $10^1$) of the PCR product of the soil samples. Amplification efficiency was 92.60%, with an $r^2$ value of 0.995.

Fungal abundance was estimated by using the ITS region of 18S as a proxy. The reaction was performed in 25.0 μL of a reaction mixture containing Power SYBR Green PCR Master Mix (Applied Biosystems, Frankfurt, Germany), 0.5 μL of each primer (100 μm), and 1.0 μL of target DNA (≈10 ng). The primers used were 5.8/ITS1F [32], and the thermal cycling conditions were as follows: 40 cycles at 94 °C for 1 min for denaturation, followed by 53 °C for 30 s for annealing and 72 °C for 1 min for final extension. The standard curve was constructed using serial dilutions ($10^6$ to $10^1$) of the PCR product of the soil samples. Amplification efficiency was 102%, with an $r^2$ value of 0.990.

Diazotrophic abundance was estimated via the *nif*H gene. The reaction was performed in 25.0 μL of a reaction mixture containing Power SYBR Green PCR Master Mix (Applied Biosystems, Frankfurt, Germany), 0.4 μL of each primer (100.0 μm), and 1.0 μL of target DNA (≈10 ng). The primers used were FPGH19/POLR [37], and the thermal cycling conditions were 40 cycles at 95 °C for 1 min for denaturation, followed by 55 °C for 27 s for annealing and 72 °C for 1 min for final extension. Amplification efficiency was 90%, with an $r^2$ of 0.9979.

Abundance of phosphate mineralizers was estimated via the *phoD* gene. The reaction was performed in 25.0 μL of a reaction mixture containing Power SYBR Green PCR Master Mix (Applied Biosystems, Frankfurt, Germany), 0.2 μL of each primer (100.0 μm), and 2.0 μL of target DNA (≈10 ng). The primers used were ALPS-F730 and ALPS-R1101 [38], and the thermal cycling conditions were 40 cycles at 95 °C for 30 s for denaturation, followed by 51 °C for 1 min for annealing and 72 °C for 30 s for final extension. Amplification efficiency was 88%, with an $r^2$ value of 1.000.

All data from the amplification of DNA were interpolated on the respective standard curve. Specificity of the amplification products was confirmed by melting curve analysis, and the expected sizes of the amplified fragments were checked on a 1% agarose gel.

### 2.2.3. Mycorrhizal Colonization (AMF)

Fine root samples of citrus plants were collected in the canopy projection and surface layer (0.0–0.10 m); in the sixth and seventh year after planting between February and March at six points, a composite sample was formed. The fine roots were clarified with immersion in KOH (10%) and then in $H_2O_2$. They were colored with Parker blue ink (s0037480) and the stained roots were immersed in lactoglycerol ($H_2O + C_3H_6O_3$ + glycerin). The percentage of colonization by arbuscular mycorrhizal fungi (AMF) was determined "via intersection of quadrants" [39]; colonies were counted on a 1.0 cm piece of root under an optical microscope.

### 2.2.4. Tahiti Acid Lime Yield (Yield)

Tahiti acid lime yield was assessed by harvesting the ripe fruits, with the standard of 90–100 g, with a minimum of 40% juice and 7 °Bx of soluble solids from four central plants from each plot, and by weighing them directly on analytical scales. This assessment was carried out between the fourth and eighth year after planting, and yield was expressed in t ha$^{-1}$.

### 2.3. Statistical Analyses

Data for biomass, soil microbial community and abundance, functional genes, and yield were analyzed by ANOVA (*F*-test) at $p < 0.05$. When the effects were significant, differences among treatments

were determined using Tukey's multiple comparison test at a significance of $p < 0.05$ through the ASSITAT® program [40].

## 3. Results

### 3.1. Biomass Yield and Deposition

*Urochloa decumbens* (DEC) produced a greater quantity of biomass (t ha$^{-1}$) in the third, fourth, and fifth years after planting in relation to *U. ruziziensis* (RUZ), and both *Urochloa* spp. showed yield decreases over time. In the second year after planting the orchard, both produced biomass values close to 10 t ha$^{-1}$, whereas in the eighth year, only 20% of this value was produced, i.e., 1.7 t ha$^{-1}$ (Figure 2A).

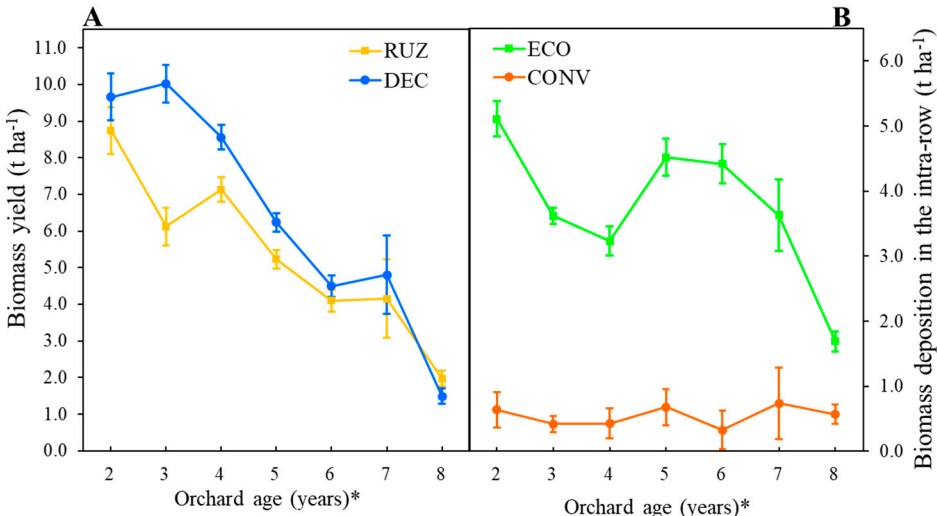

**Figure 2.** Yield (**A**) and deposition (**B**) of biomass from *Urochloa* spp. in the intra-rows, tested with different cover crops, mowers, and the use of glyphosate. The figures represent the sum of three mowings per growing season (Mogi Mirim, São Paulo, 2012–2019). Legend: RUZ = *Urochloa ruziziensis*, DEC = *U. decumbens*, ECO = ecological mower, CONV = conventional mower. The significant difference was accounted for by the Tukey test ($p < 0.05$), and the error bars represent the minimal significant difference. * Orchard age: 2012–2013 (2), 2013–2014 (3), 2014–2015 (4), 2015–2016 (5) 2016–2017 (6), 2017–2018 (7), and 2018–2019 (8).

The use of the ecological mower (ECO) resulted in a greater deposition of biomass in the intra-row when compared to the conventional mower (CONV) in all evaluation years (Figure 2B). The values of cover crop biomass were, on average, seven times higher for the ECO compared to the CONV, with the greatest difference observed between the values in the second year of the orchard.

### 3.2. Soil Microbial Activity and Abundance

Basal soil respiration—BSR (Figure 3A), differed in the fourth year after planting from the citrus tree, with the highest observed value observed in the presence of *Urochloa decumbens* and in interaction with ecological mower (Figure S4A). In the remaining years, BSR showed no difference between cover crops, with values close to 1.1 mg $CO_2$ g soil$^{-1}$. The use of ECO resulted in a higher level of soil respiration, with increments of 78%, 81%, 25%, and 31%, respectively, in the years evaluated, in relation to the use of the conventional mower. The absence of glyphosate (NH) favored BSR, with significant differences in the seventh and ninth year after planting, with an increase of 7% and 25%, respectively, relative to the herbicide treatment. In interaction with ECO, the NH produced a similar result, with an increase of 15% and 36%, respectively, relative to the use of glyphosate (H), in both years (Figure S4A).

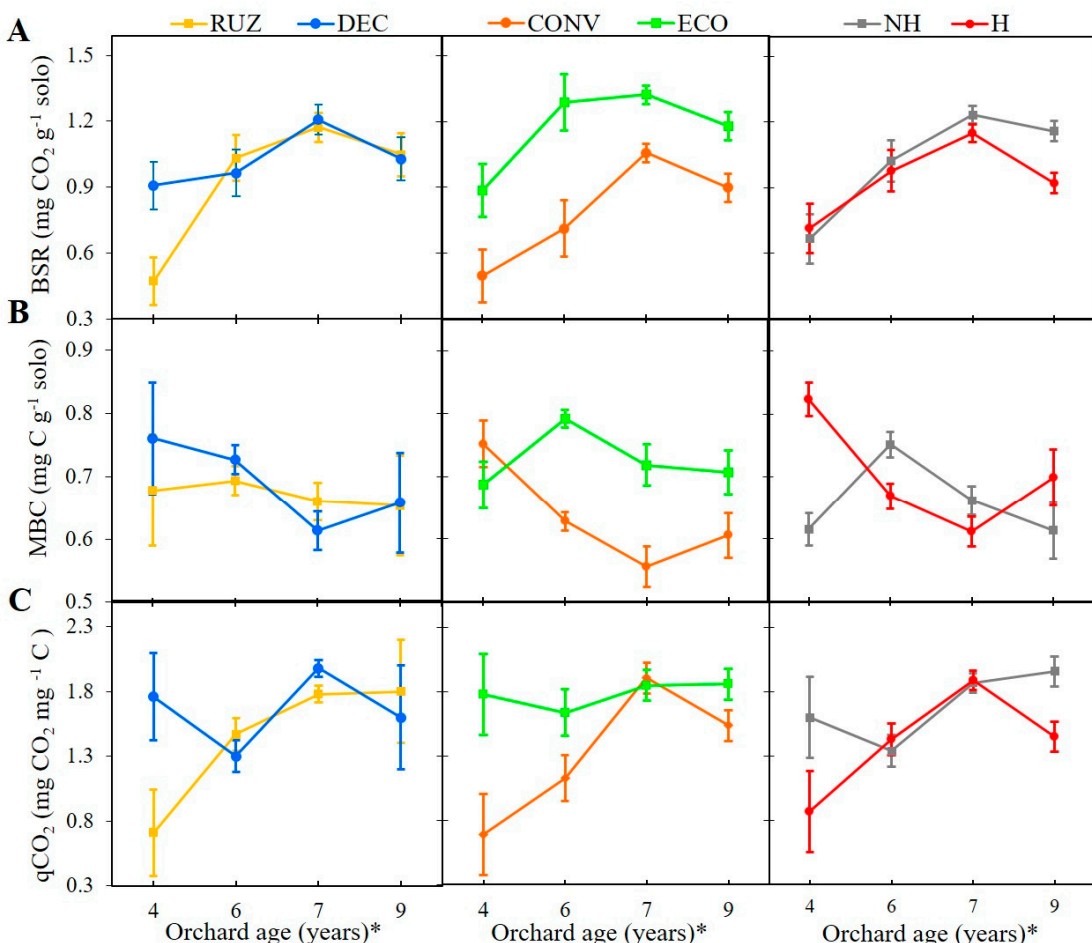

**Figure 3.** Basal soil respiration, BSR (**A**), microbial biomass carbon, MBC (**B**), and metabolic quotient, qCO$_2$ (**C**), tested with different cover crops, mowers, and the use of glyphosate (Mogi Mirim, São Paulo, 2014–2019). Legend: RUZ = *Urochloa ruziziensis*, DEC = *U. decumbens*, ECO = ecological mower, CONV = conventional mower, H = glyphosate herbicide (1080 g ae. ha$^{-1}$), NH = no herbicide. The significant difference was accounted for by the Tukey test ($p < 0.05$), and the error bars represent the minimal significant difference. * Orchard age: growing season 2014–2015 (4), 2016–2017 (6), 2017–2018 (7), and 2019–2020 (9).

The *Urochloa* species did not shown significant difference between them ($p > 0.05$) as to microbial biomass carbon (MBC) (Figure 3B). However, the use of ECO increased the amount of microbial biomass carbon relative to the use of CONV in the sixth, seventh, and ninth years after planting. The RUZ in interaction with ECO increased in ninth year, with 0.75 mg C g$^{-1}$ (Figure S4B). In the sixth and seventh years after planting, the absence of glyphosate facilitated MBC growth (Figure 3B).

The used of DEC resulted in a greater metabolic quotient (qCO$_2$) in the fourth year after planting in relation to RUZ (Figure 3C). ECO provided a higher qCO$_2$ value than that of CONV in the fourth, sixth, and ninth years after planting. A high qCO$_2$ value was also found in the interaction of DEC with ECO or NH in the fourth year (Figure S4C). The NH treatment had a greater qCO$_2$ in the fourth and ninth years (Figure 3C) and in interaction with ECO (Figure S4C).

Regarding microbial abundance, the presence of *Urochloa ruziziensis* in the citrus inter-row increased the fungal (ITS) and bacterial (16S) abundances in the soil by 13.0% and 1.4%, respectively, relative to *U. decumbens*, in the seventh year after planting (Figure 4A,B). In the sixth year after planting, the ecological mower increased microbial abundance by 8.2% and 3.3%, respectively, in relation to conventional mower. Herbicide treatment did not influence the abundances of fungi and bacteria (Figure 4A,B).

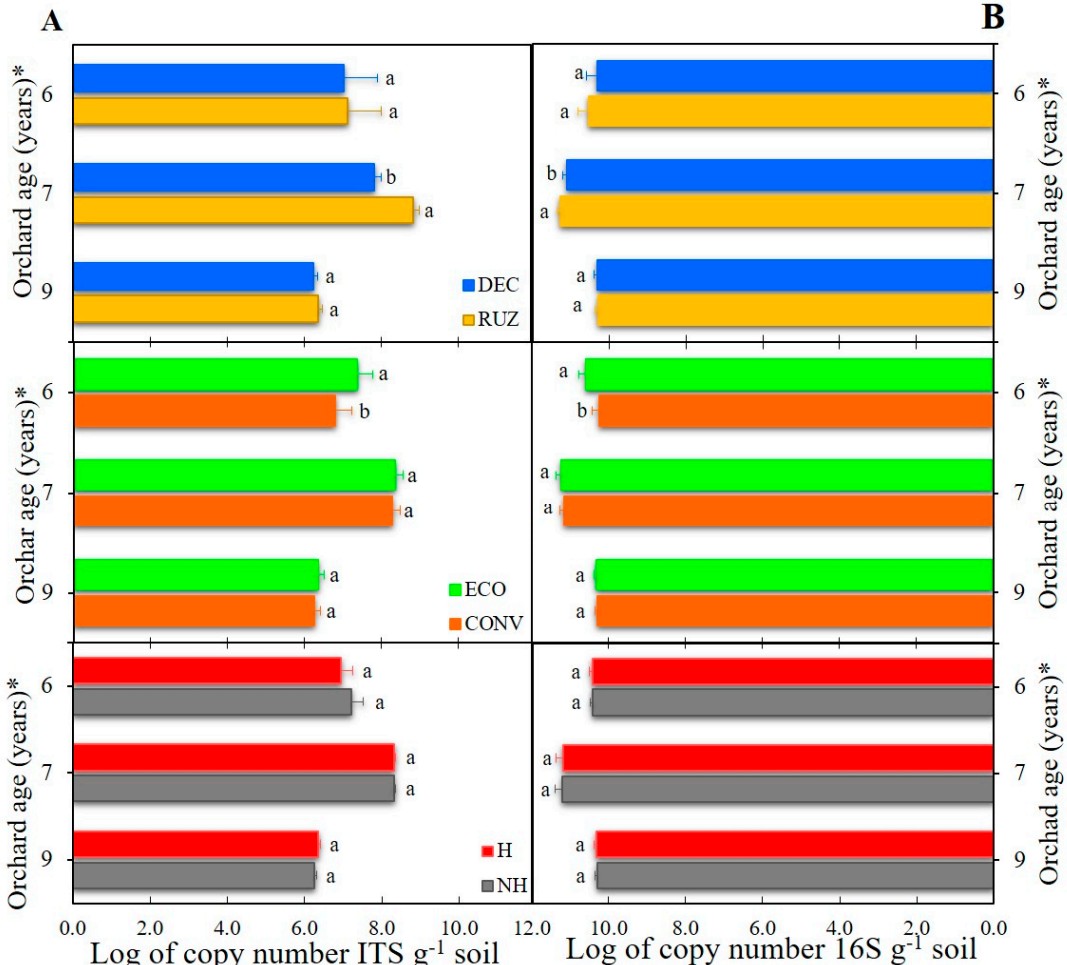

**Figure 4.** Abundance of the fungal gene ITS (**A**) and bacterial 16S RNA (**B**) from the rhizosphere of Tahiti acid lime plants tested with different cover crops, mowers, and the use of glyphosate (Mogi Mirim, São Paulo, 2016–2019). Legend: RUZ = *Urochloa ruziziensis*, DEC = *U. decumbens*, ECO = ecological mower, CONV = conventional mower, H = glyphosate herbicide (1080 g ae. ha$^{-1}$), NH = no herbicide. The significant difference was accounted for by the Tukey test ($p < 0.05$), and the error bars represent the minimal significant difference. * Orchard age: growing season 2016–2017 (6), 2017–2018 (7), and 2019–2020 (9).

The *U. ruziziensis* and ecological mower treatments increased the abundances of the functional genes *nif*H and *pho*D relative to the *U. decumbens* and conventional mower treatments, respectively (Figure 5A,B). The abundances of the *nif*H gene and *pho*D were 7.18 and 8.16 copies (log) of the gene g soil$^{-1}$, with RUZ, respectively, and 7.22 and 8.21 copies (log) of the gene g soil$^{-1}$ with ECO, of the *nif*H and *pho*D, respectively. There was no difference between treatments with or without glyphosate (Figure 5A,B). DEC with CONV may have damaged the abundance of the 16S and *pho*D gene, as the lowest values were obtained in the sixth and seventh years (Figure S5A,B).

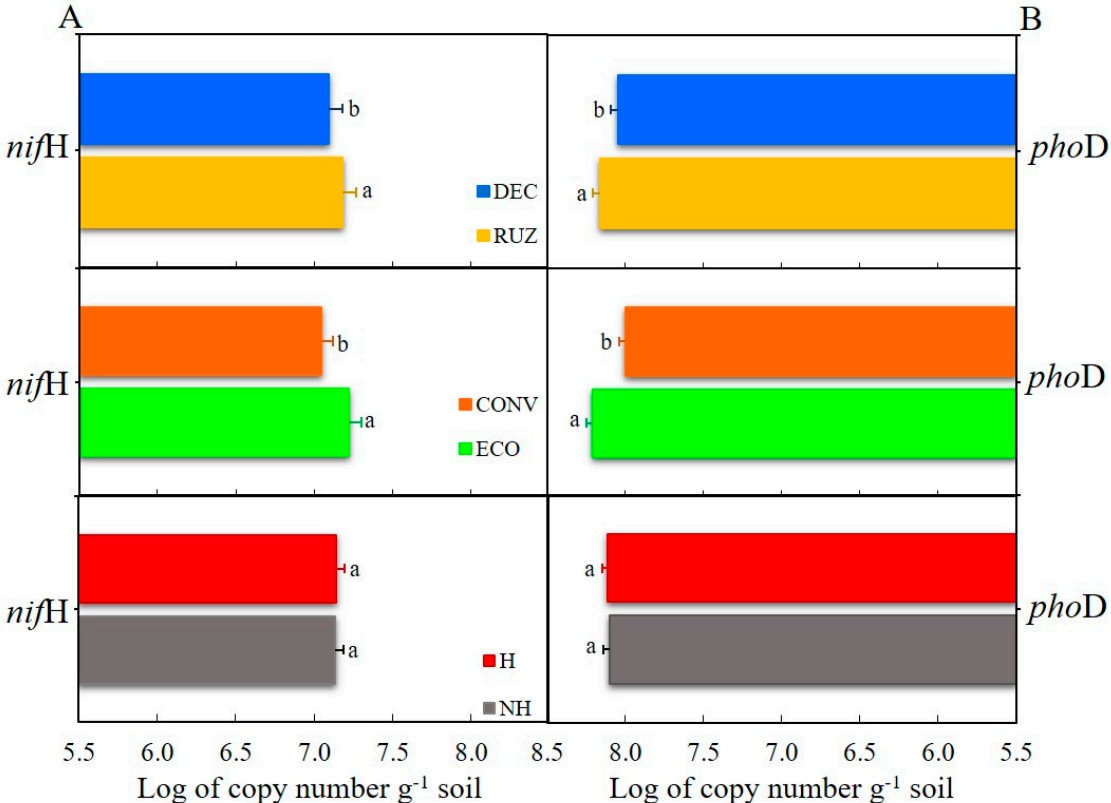

**Figure 5.** Abundance of the nitrogen fixer enzyme gene, *nif*H (**A**) and alkaline phosphatase, *pho*D (**B**) from the rhizosphere of Tahiti acid lime plants tested with different cover crops, mowers, and the use of glyphosate (Mogi Mirim, São Paulo, 2019). Legend: RUZ = *Urochloa ruziziensis*, DEC = *U. decumbens*, ECO = ecological mower, CONV = conventional mower, H = glyphosate herbicide (1080 g ae. ha$^{-1}$), and NH = no herbicide. The significant difference was accounted for by the Tukey test ($p < 0.05$), and the error bars represent the minimal significant difference.

### 3.3. Mycorrhizal Colonization (AMF)

The use of *U. ruziensis* resulted in an increase of 34.7% in the colonization of the roots by arbuscular mycorrhizae fungi (AMF) in relation to the use of *U. decumbens* in the sixth year after planting (Figure 6). In the sixth and seventh years, the increase was 44.5% and 17.8%, respectively, for ecological mower treatments compared to use of a conventional mower. The use of glyphosate had no impact on mycorrhizal colonization (Figure 6). In interaction with RUZ and ECO, they brought about greater colonization (Figure S6).

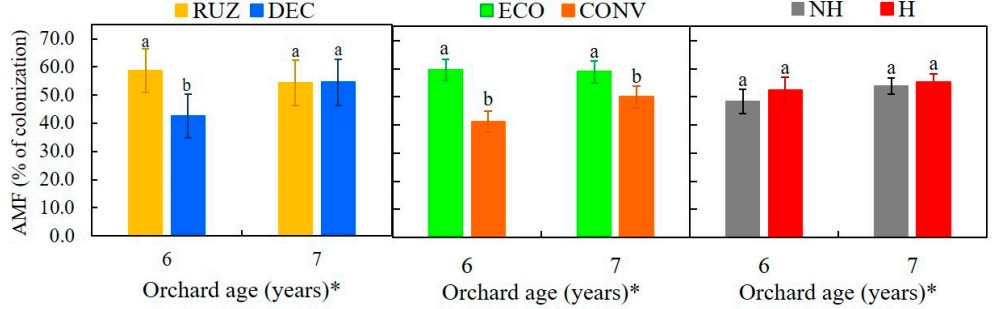

**Figure 6.** Percentage of colonization by arbuscular mycorrhizae fungal (AMF), tested with different cover crops, mowers, and the use of glyphosate (Mogi Mirim, São Paulo, 2016–2017). Legend: RUZ = *Urochloa ruziziensis*, DEC = *U. decumbens*, ECO = ecological mower, CONV = conventional mower, H = glyphosate herbicide (1080 g ae. ha$^{-1}$), NH = no herbicide. The significant difference was accounted for by the Tukey test ($p < 0.05$), and the error bars represent the minimal significant difference. * Orchard age: growing season 2016–2017 (6) and 2017–2018 (7).

### 3.4. Tahiti Acid Lime Fruit Yield (Yield)

The highest fruit yield was achieved with *Urochloa ruziziensis* treatment in the eighth and ninth years after planting compared with *U. decumbens* (Figure 7). Maintaining soil cover in the intra-row using the ecological mower resulted in a greater yield from the fourth to the eighth year, compared with the conventional mower (Figure 7). From the fourth to the sixth year after planting, the yield was higher in the use glyphosate treatment compared to the non-glyphosate treatment.

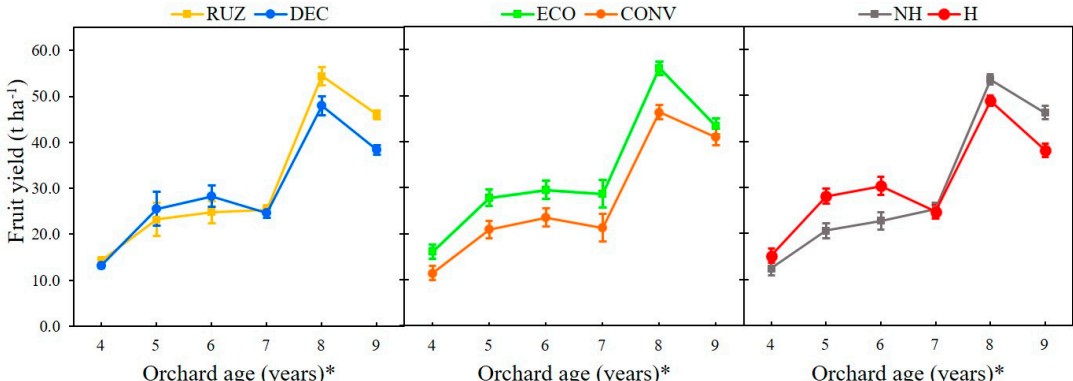

**Figure 7.** Orchard fruit yields for different cover crops, mowers, and the use of glyphosate (Mogi Mirim, São Paulo, 2014–2019). Legend: RUZ = *Urochloa ruziziensis*, DEC = *U. decumbens*, ECO = ecological mower, CONV = conventional mower, H = glyphosate herbicide (1080 g ae. ha$^{-1}$), NH = no herbicide. The significant difference was accounted for by the Tukey test ($p < 0.05$), and the error bars represent the minimal significant difference. * Orchard age: growing season 2014–2015 (4), 2015–2016 (5), 2016–2017 (6), 2017–2018 (7), and 2019–2020 (9).

The use of the herbicide with RUZ increased the yield in the fourth year. From the seventh year on, the interaction of *Urochloa* spp. and the mowers showed an increase in yield in the absence of glyphosate (Figure S7).

### 3.5. Correlation between the Microbial Community and Environmental Variables

The redundancy analysis (RDA) correlated the environmental variables of basal soil respiration, microbial biomass carbon, metabolic quotient, mycorrhizal colonization, abundance of genes 16S, ITS, *nif*H, *pho*D, and the yield of Tahiti acid limes with the bacterial (Figure 8A–C) and fungal (Figure 9A–C) community in the different treatments.

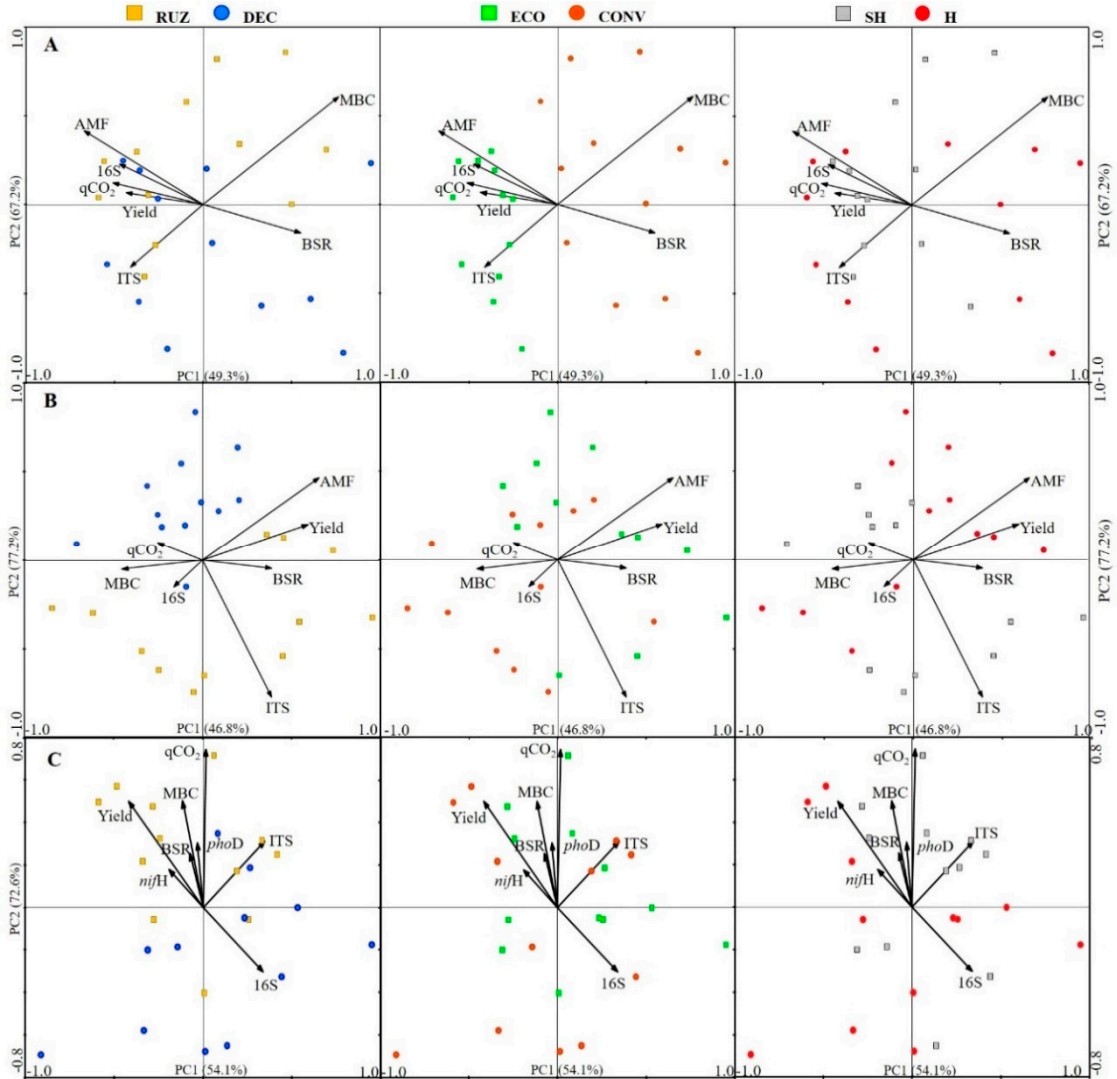

**Figure 8.** Redundancy analysis based on biplot distance from the bacterial community (Bray–Curtis) in a Tahiti acid lime orchard, under testing with different cover crops, mowers, and the use of glyphosate (Mogi Mirim, São Paulo, 2016–2019). Legend: RUZ = *Urochloa ruziziensis* (yellow squares), DEC = *U. decumbens* (blue circles), ECO = ecological mower (green squares), CONV = conventional mower (orange circles), H = glyphosate herbicide (1080 g ae. ha$^{-1}$) (red circles), NH = no herbicide (gray squares). Orchard age: growing season, 2016–2017 (**A**), 2017–2018 (**B**) and 2019–2020 (**C**).

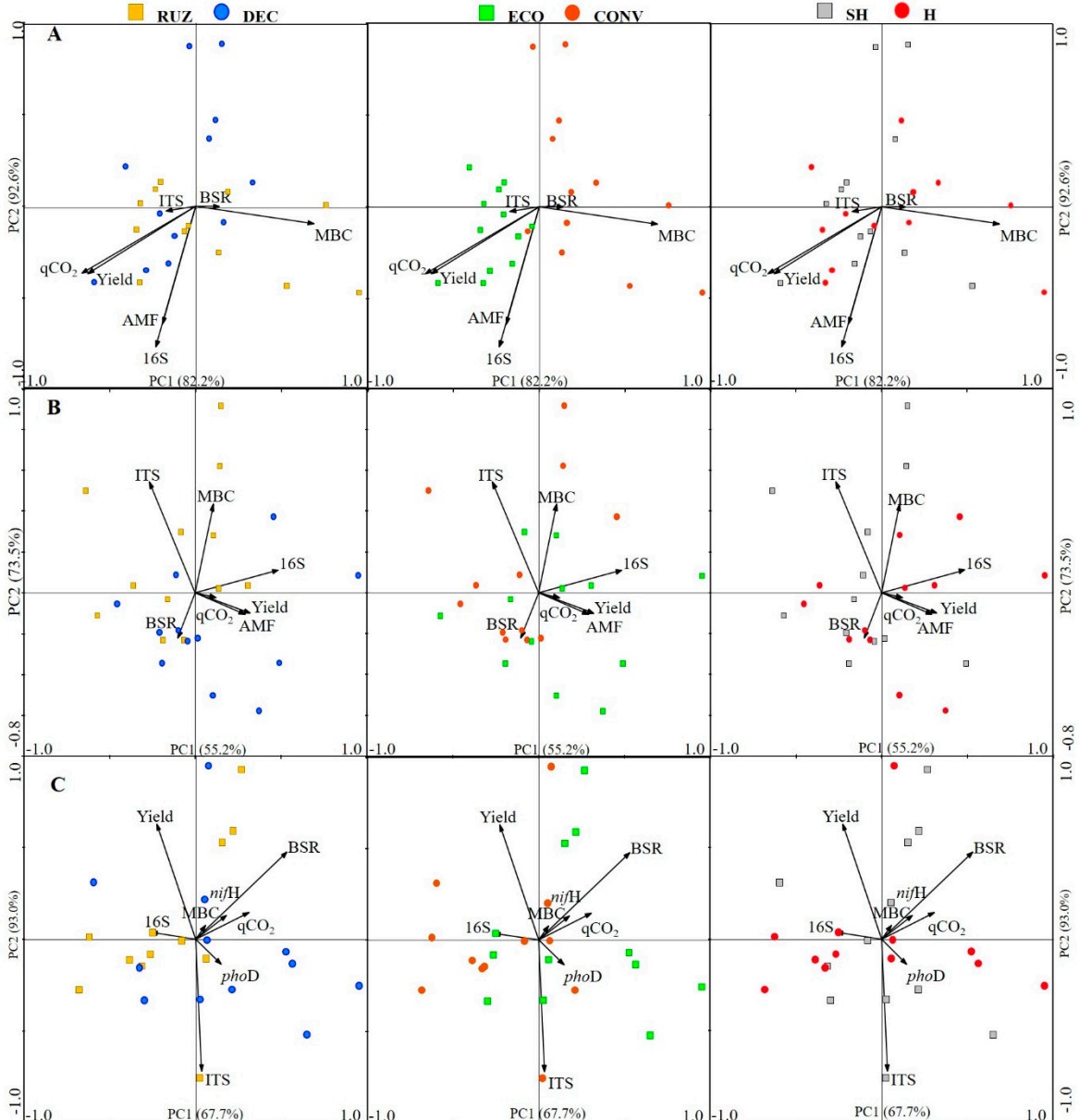

**Figure 9.** Redundancy analysis based on biplot distance from the fungal community (Bray–Curtis) in a Tahiti acid lime orchard, under testing with different cover crops, mowers, and the use of glyphosate (Mogi Mirim, São Paulo, 2016–2019). Legend: RUZ = *Urochloa ruziziensis* (yellow squares), DEC = *U. decumbens* (blue circles), ECO = ecological mower (green squares), CONV = conventional mower (orange circles), H = glyphosate herbicide (1080 g ae. ha$^{-1}$) (red circles), NH = no herbicide (gray squares). Orchard age: growing season, 2016–2017 (**A**), 2017–2018 (**B**) and 2019–2020 (**C**).

In all RDA with the environmental variables and soil bacterial community (Figure 8A–C), principal component 2 (PC2) presented greater statistical information (67.2%, 77.2%, 72.6%) in relation to principal component 1 (PC1), which presented 49.3%, 46.8%, 54.1%, in all evaluated years (sixth, seventh, and ninth).

In the first evaluation (Figure 8A), the variables of Yield, AMF, and MBC showed significant differences ($p < 0.05$) in relation to the other environmental variables, with the distribution of the bacterial community in the soil. The microbial biomass carbon variable was the one with the highest correlation in relation to the distribution of the bacterial community, presented by the largest size of the vector; however, it is not related to the ITS gene, due to the opposite position of the vector.

The treatments with *Urochloa* spp. and herbicide had no effect on this distribution, since there is no distinction in the bacterial community. However, the bacterial community with the use of the ecological mower correlated positively with most variables (AMF, 16S, $qCO_2$, and yield), which are more associated with ITS, compared to MBC and BSR, and which showed a positive correlation with the bacterial community when using the conventional mower.

In the second evaluation (Figure 8B), the AMF and ITS variables showed the highest and most significant correlations ($p < 0.05$) in relation to the other variables and the distribution of the bacterial community in the soil; however, they have little relation to each other. Furthermore, despite not showing any significance, yield showed a strong correlation. It is possible to observe that the vectors AMF and 16S are positioned in the opposite way. The treatments with the ecological mower and the herbicide had no effect on the distribution of the bacterial community. However, *Urochloa* spp. formed two different groups, with only the $qCO_2$ variable showing a positive correlation to DEC, with the other variables being correlated with *Urochloa ruziziensis*.

In the third evaluation (Figure 8C) there was no significant difference between the variables, however the $qCO_2$ and the yield showed a greater correlation, displaying a greater influence on the distribution of the bacterial community, as presented by the larger size of the vectors. All variables except 16S were observed to be associated. Regarding the treatments, there was a certain differentiation of the community with the use of different species of *Urochloa*, with the majority of the environmental variables (yield, *nif*H, $qCO_2$, BSR, MBC, *pho*D, and ITS) positively correlating with the distribution of the bacterial community of RUZ.

In the analysis of RDA with the environmental variables and soil fungal community (Figure 9A–C), the axis of principal component 2 (PC2) presented more statistical information (92.6%, 73.5%, 93.0%) compared to PC1, which presented 82.2%, 55.2%, 67.7% in all evaluated years (sixth, seventh, and ninth).

In the first evaluation (Figure 9A), a greater correlation was found between the distribution of the fungal community with the variables $qCO_2$, AMF, 16S, yield, and MBC, but this was only significant ($p < 0.05$) for the last two. The BSR variables with MBC and ITS, $qCO_2$, yield, and AMF with 16S were related, forming two opposite groups. Regarding treatments, the use of different species of *Urochloa*, and the herbicide, the distribution of the community was dispersed, making it impossible to ascertain a correlation. However, two groups were found in relation to the mower type, with a positive effect of the ECO grouping with the variables ITS, $qCO_2$, yield, AMF, and 16S and the CONV being grouped with BSR and MBC.

In the second evaluation (Figure 9B) the greatest correlations in the distribution of the fungal community were with the variables ITS, MBC, and 16S, but without significance ($p > 0.05$). It is possible to verify the greater association between yield, AMF, and $qCO_2$, and the opposite relationship with ITS. Regarding treatments, the variables BSR, $qCO_2$, AMF, and yield have a certain correlation with the distribution of the fungal community in the DEC. The other treatments did not show a dispersion of the community that could be correlated with any of the variables.

In the third assessment (Figure 9C), the BSR, ITS, and yield variables showed the highest correlation, but there were no significant differences ($p > 0.05$) between them and the distribution of the fungal community. The variables MBC, BSR, *nif*H, and $qCO_2$ were associated; however, yield displayed an opposite relationship with *pho*D, which is associated with ITS. *Urochloa* spp. and mower type showed some differentiation in the distribution of the fungal community, with a positive correlation of the variables *pho*D and ITS with RUZ and ECO.

## 4. Discussion

### 4.1. Biomass Yield and Deposition

The mean *Urochloa decumbens* biomass yield observed between the second and fourth years of tree planting in the field was 7.3 t ha$^{-1}$ after three mowing events per year. A similar result was observed for the *U. decumbens* biomass yield (6.4 t ha$^{-1}$), in a single cut, in the inter-row of a young

Pera orange orchard in the state of Amazonas (Lat 3′25″ S), Brazil [41]. This region shows higher levels of rainfall and as well as air temperatures, compared with the state of São Paulo (Lat 22′50″ S). The yield obtained in Amazonas shows the productive potential of this species under conditions that are beneficial for growth.

The decrease in the incidence of solar radiation in the inter-rows and machinery traffic hampered the full development of plants of *Urochloa* spp., with a decrease in the production of biomass over time. Heavy machine traffic destroys the soil structure and decreases soil biological quality. The destruction of macroaggregates, which are used as habitats of the microbiota, results in the loss of biomass of the cover crops [10,42].

The correct management of the cover crops, with the deposition of their biomass to the citrus intra-rows, using the ecological mower, may contribute to increased biological activity of the soil, as well as a higher nutrient availability due to the decomposition of the vegetable residues and an increase in the organic matter in the soil [10]. Covering of the soil with biomass, which is common in no-till systems, also contributes to the maintenance of soil moisture and aggregate stability, as well as increasing organic matter and consequently, increase crop yield [43].

The use of grasses in intercropping with citrus plants is advantageous, considering that they are plants with perennial cover crops, with production of biomass and high ratios of C:N, with an average value of 46 in southeastern Brazil [44]. This can facilitate soil microbial growth and carbon sequestration by increasing the labile carbon in the soil, thus facilitating the increase in soil organic matter (SOM) [42]. The increase in SOM in the intra-row is of particular importance in sandy soils, where the low clay content leads to a lower retention of the organic material in colloids, thereby reducing losses caused by the oxidation of SOM and carbon, in the form of $CO_2$ [18].

### 4.2. Soil Microbial Activity

The emission of $CO_2$ from the soil is associated with microbial activity, root respiration, decomposition of vegetable residues, and oxidation of soil organic matter [45]. The largest basal soil respiration in the ecological mower treatment, relative to the conventional mower, is associated with the intake of biomass from *Urochloa* spp. in the intra-row, indicating high microbial activity. The degradation of this residue by microorganisms for use as a source of energy releases newly immobilized nutrients to plants [42,46]. The effects of sugarcane straw mulching have been studied, and a higher rate of basal soil respiration was observed, which is a sensitive indicator of disturbances in soil microbial communities in the covered soil compared to the uncovered soil, favoring the intensive consumption of oxidable carbon by the microbial population [47].

Mulching with *Urochloa* spp. biomass on the intra-row (ECO) reduced the temperature amplitude and maintained the moisture in the soil, resulting in higher microbial activity. In organic orange production systems, where organic fertilizers are used and the soil is covered, higher microbial activities have been reported [10]. The decrease in BSR in the herbicide treatment compared to the non-herbicide use in the seventh and ninth years after planting may be associated with changes in the microbial community, such as an altered fungal abundance [12,15]. Previous studies have shown little interference of herbicides with the microbial community, and larger BSR values have been found due to the degradation of glyphosate [15,16,48].

The microbial biomass carbon level was higher for the ecological mower treatment than for the conventional mower treatment, due to the constant maintenance of soil cover with biomass from *Urochloa* spp. The microorganisms use SOM as a source of nutrients and energy for their development, temporarily immobilizing these resources. After cell death, these resources are released, reducing the loss of nutrients in the soil-plant system [46,49]. *U. humidicola* intercropped with young Pera orange (4 years), installed via minimum tillage, resulted in 0.27 mg C g soil$^{-1}$, which may be associated with characteristics of the species and the non-deposition of the residues in the intra-row [42]. The MBC value found in coffee intercropped with *U. decumbens* in the inter-row was 0.6 mg C g soil$^{-1}$ [46]. An MBC of 0.16 mg C g soil$^{-1}$ was verified with the use of wheat straw as mulch [50]. The use of Poaceae

species, which show a higher C:N (carbon/nitrogen) ratio in relation to Fabaceae, can help to accumulate atmospheric carbon due to their greater carbon absorption efficiency. Thus, soil microbiological activity is favored by the quality and quantity of C-based inputs allocated to the soil [42].

The microbial biomass, despite being a source of C reserve, transforms other nutrients, such as N and P, which are important in nutrient cycling [49], and accounts for about 5% of the SOM. The active part is more labile compared to the non-living part of organic matter (95%) [42]. The microbiota can be used as a soil biological quality index, responding rapidly to system changes [12] due to their dynamic relationship with the quality and quantity of the mulch.

The largest metabolic quotient values in the fourth, sixth, and ninth years after planting in the ecological mower treatment can be associated with the largest energy expenditure in the microbial community, to maintain microbial activities, as well as with the presence of more active microorganisms [46]. A similar situation was found in soybean planting, where the lack of milliliter mulch resulted in a greater $qCO_2$ (3.0 mg $CO_2$ g $C^{-1}$ $h^{-1}$) [49]. Thus, the increase in the coefficient generates a greater loss of carbon via $CO_2$ and a lower carbon sequestration rate.

The abundance of microorganisms in the soil is related to the presence of plant residues in the soil. The soil microbial biomass is made up mainly of fungal and bacterial species [49]. Greater quantities of biomass from *Urochloa* spp. plants were deposited in the citrus intra-row in the ecological mower treatment in the second, fifth, and sixth years. The abundances of fungi and bacteria were higher in the sixth and seventh years after planting, with differences between mowers and *Urochloa* spp., respectively.

The use of *Urochloa ruziziensis* as cover crop facilitated the fungal and bacterial community in the seventh year after planting, possibly because of the root exudates and the straw released to the soil. Root exudates can facilitate the diversity and abundance of soil microbiota due to their carbon compounds, which can easily be degraded and serve as a source of energy and nutrients for microbial metabolism [42]. The ecological mower treatment provided a greater fungal and bacterial abundance in the sixth year after planting, corroborating the data from [50], in which higher bacterial abundances were found in the soil of strawberry plants growing under wheat straw.

To evaluate the biological quality of the soil, the presence of functional genes of some microorganisms that promote plant growth can be measured. The *nif*H gene, which indicates the presence of nitrogen-fixing bacteria (diazotrophic bacteria) [12], and the *pho*D gene, which assists in the release of phosphorus [51], were more abundant in ecological mower and *Urochloa ruziziensis* treatments in the ninth year after planting. Nitrogen is an essential element for plant development and is dependent on the microbial community of the soil. Nitrogen mineralization and immobilization are associated with microbial biomass, functional genes, and the carbon and nitrogen cycling rates of microorganisms [12]. Systems that improve the proportion of these factors may be essential for the good development and yield of Tahiti acid lime.

The community of nitrogen-fixing bacteria benefits from organic fertilization, minimum soil tillage, and greater soil humidity [12]. Researchers observed no increase in nitrogen cycling bacteria in an organic orange orchard with *Brassica* ssp. or uncovered soil [52]. This shows that the management and quality of the residue influence the nitrogen-fixing bacteria.

Bacteria containing the *pho*D gene, via alkaline phosphomonoesterase (ALP) enzymes, are important for organic phosphorus mineralization [51]. A low availability of inorganic P assists in the activation of these enzymes. With increasing fertilization with inorganic P, the enzyme activity and abundance of the *pho*D gene are lowered [53]. The higher abundance of the *pho*D gene in *Urochloa ruziziensis* and ecological mower is associated with the species of the cover crop and with the deposition of the straw in the intra-row.

In soils with added plant residues, a larger quantity of microorganisms with specific functions, i.e., degradation functions, can be found due to the presence of the residue itself. This is in contrast to systems without the addition of plant residues (conventional mowing), where possibly more microorganisms with more dispersed and non-specific functions [54] are present. Thus, systems with

lower microbial diversity influence nutrient cycles, which can impair crop yields due to the lower supplies of C, N, and P [55].

Due to the importance of the study of the soil microbial community, future research should provide for the metagenome analysis of soil bacteria and fungi, also identifying specific genes and enzymatic activities of the soil microbial community of importance for the cultivation of citrus and soil quality.

### 4.3. Mycorrhizal Colonization (AMF)

Arbuscular mycorrhizae fungi facilitate carbon sequestration and increase the presence of organic carbon in the soil, transfer assimilated carbon to the plants by symbiosis [56], improve nutrient and water absorption, and increase soil quality and aggregation [57,58]. In the mulched intra-row (ECO treatment), the percentage of root colonization by arbuscular mycorrhizae fungi was increased. Mulching with plant biomass and the absence of herbicide can promote a larger network of AMF mycelium, which can contribute to soil aggregation, as well as to higher infiltration and aeration rates [58].

### 4.4. Correlation between the Microbial Community and Environmental Variables

The redundancy analysis of the soil bacterial community in the sixth year (Figure 8A) showed that the environmental variables (mycorrhizal colonization, abundance of the gene 16S, metabolic quotient and yield) are related and have less association with abundance of the gene ITS. This relationship may indicate that these variables are providing better quality in the soil, with effects on Tahiti acid lime yield. It is possible to observe the correlation of these variables with the bacterial community formed by the ecological mower, which provides benefits for the deposition of the biomass of *Urochloa* spp. in the inter-row area. The soil bacterial community formed by the conventional mower, which deposits biomass in the intra-rows, showed a greater correlation with microbial biomass carbon and basal soil respiration, which had little association (Figure 8A–C). This can be justified by the fact that, in the same year, these variables showed different responses, with an increase in BSR (Figure 3A) and a decrease in MBC (Figure 3C).

A result similar to the bacterial community was found for the distribution of the fungal community in the same evaluation period (Figure 9A). A positive correlation was observed between the community formed by ECO and the variables of AMF, 16S, $qCO_2$ and yield. A positive correlation was also observed between the conventional mower and the BSR and MBC variables; that is, the fungal community of the CONV is defined by the microbial activity and the carbon in the biomass.

In the second assessment (Figure 8B) the RDA of the bacterial community with the environmental variables formed two groups (AMF, yield, BSR, and ITS on the one hand; and $qCO_2$, MBC, and 16S on the other), but with little relation to each other or correlation with the formed communities, demonstrating the non-response of these variables regarding the treatments (Figure 3A–C, Figure 4A,B, Figure 6, and Figure 7). The ITS variable showed a significant positive correlation with the bacterial community of RUZ. We also found greater quantification of this gene with the use of *U. ruziziensis*, which was equal in biomass yield to *U. decumbens* in the seventh year (Figure 2A).

A similar result was found for the fungal community, as shown in the RDA (Figure 9B), with the variable ITS having a positive correlation with RUZ. However, the variables $qCO_2$, yield, and AMF, which were associated and which had little positive correlation with the fungal community of ECO, responded positively to the treatment with the ecological mower, as they did for yield (Figure 7) and AMF (Figure 6). Previous researchers [58] found a positive correlation between the increased diversity of AMF species and the use of *P. notatum* as a cover crop for citrus orchards.

Most of the variables were associated, except 16S (Figure 8C), which did not obtain a significant difference regarding the treatments in the ninth year (Figure 4A). Although this variable (16S) symbolizes the quantification of bacterial genes, it did not present a significant correlation with the structure of the soil bacterial community. Despite the low straw yield by both species of *Urochloa* (Figure 2A), there was

a distinction between the bacterial community and the type of cover crop, with a positive correlation of most variables with RUZ, such as yield, with a positive effect for RUZ's bacterial community, and high value in the treatment with *U. ruziziensis* (Figure 7). A similar result was found in the distribution of the fungal community, with a positive correlation in the community with the use of *U. ruziziensis* and the yield variable (Figure 9C). The *pho*D gene showed greater quantification with the use of *U. ruziziensis* (Figure 5B) and a positive correlation with the soil bacterial community, different from the fungal community, which displayed a positive correlation with DEC. This relationship can be associated with the presence of this gene in bacterial communities [51].

The herbicide treatment (SH or H) did not influence the soil bacterial or fungal community in the last years, and was no correlation with the analyzed variables. This demonstrates that in an adult Tahiti acid lime orchard, herbicide does not influence soil microbiology and its non-use will not cause damage to the orchard.

Different responses can be found from one year to another, since the cover crops decreased the production of biomass and there was also a drop in the deposition of this biomass, mainly in the line (ECO), as shown in Figure 2. This fact shows that different management practices influence the soil microbial community [59].

### 4.5. Soil Microbiological Quality

The maintenance of plant residues in soil, combined with minimum tillage, results in improved soil chemical, physical, and microbiological attributes, which can improve soil quality [10,12].

In perennial crop systems, such as citrus orchards, alternatives should be adopted to improve the biological quality of the soil, such as crop rotation and the use of cover crops. While crop rotation is impractical in perennial crops, mulching with *Urochloa* spp. in the inter-rows and launching it into the rows (intra-rows) can be an alternative method of increasing microbiological quality, making the system more sustainable and productive. According to [18], to improve sustainability, it is essential to maintain soil organic matter as a source of nutrients and energy for microorganisms through cover crops such as Fabaceae and Poaceae species.

The use of intercropping, as well as the depositing of the biomass into the intra-rows, as proposed in this work, may result in certain benefits to the orchard. Cover crops should, however, not compete for water and nutrients, they should protect the attributes of the soil, resulting in greater biological activity, greater availability of nutrients, and protection against destructive processes, which is reflected in higher yields [42].

The use of herbicides has resulted in increased yields up to the sixth year after planting. Weed control is important in young orchards where canopy shadowing is low and weeds are not naturally suppressed. Furthermore, competition for water and nutrients is high during this period [13]. The depositing of *Urochloa* spp. biomass in the intra-row (the ecological mower method) had a positive effect on the soil microbiological attributes, leading to a greater than 20% increase in the average yield of Tahiti acid limes when compared to the treatment without mulching, which is equal to 36 t ha$^{-1}$, higher than the average, 26.7 t ha$^{-1}$ yield for crops in Brazil [20].

Based on our results, *Urochloa ruziziensis* was the best cover crop for the intercropping of a citrus orchard and keeping the soil covered with straw with an ecological mower improved soil microbial activity and abundance, as well as soil genetic functionality. Soil microbiology was influenced little by the use of herbicide.

Future work should be carried out with other citrus species and including measurements of the soil's physical quality. This form of crop management can be used by small and large growers, resulting in a better quality production system and higher income.

## 5. Conclusions

The use of *Urochloa ruziziensis* and an ecological mower, independent of the use of herbicide, improved soil microbiological quality and increased Tahiti acid lime yield.

**Supplementary Materials:** The following are available online at http://www.mdpi.com/2077-0472/10/11/491/s1, Figure S1. Number of peaks and standard deviation (SD) obtained for biological community evaluated in different cover crops and mowers (Mogi Mirim-SP, 2016–2019), Figure S2. Number of peaks and standard deviation (SD) obtained for fungal community evaluated in different cover crops and mowers (Mogi Mirim-SP, 2016–2019), Figure S3. Sequence of oligonucleotides used for gene quantification, Figure S4. Two-way interactions of basal soil respiration (**A**), microbial biomass carbon (**B**), and metabolic quotient (**C**), tested with different cover crops, mowers, and the use of glyphosate (Mogi Mirim, São Paulo, 2014–2019), Figure S5. Two-way interactions of abundance of the bacterial gene, 16S (**A**) and alkaline phosphatase, *pho*D (**B**) tested with different cover crops and mowers (Mogi Mirim, São Paulo, 2016–2019), Figure S6. Two-way interactions of colonization by arbuscular mycorrhizae fungi, tested with different cover crops and mowers (Mogi Mirim, São Paulo, 2016–2017), Figure S7. Two-way interactions of Tahiti acid lime fruit yield with different cover crops, mowers, and the use of glyphosate (Mogi Mirim, São Paulo, 2014–2019).

**Author Contributions:** Conceptualization, A.C.C.A., and F.A.d.A.; methodology, A.C.C.A., D.M.-J., and F.A.d.A.; software, S.R.C. and R.M.; validation, P.M.d.C., R.M.B., F.D.A., and D.M.-J.; formal analysis, A.C.C.A, S.R.C., R.M., and F.A.d.A.; investigation, A.C.C.A., R.M., and A.G.P.; resources, S.P.M., F.D.A., and D.M.-J.; writing—original draft preparation, A.C.C.A.; writing—review and editing, A.C.C.A., S.R.C., P.M.d.C., F.D.A., D.M.-J., and F.A.d.A.; visualization, A.C.C.A.; supervision, F.A.d.A.; project administration, A.C.C.A., R.M., and F.A.d.A.; funding acquisition, D.M.-J. All authors have read and agreed to the published version of the manuscript.

**Funding:** This research was funded by Fundação de Amparo à Pesquisa do Estado de São Paulo (FAPESP, Brazil), grant number #2014/21349-4, Conselho Nacional de Desenvolvimento Científico e Tecnológico (CNPq, Brazil), grant number #408362/2016-2, and Instituto Nacional de Ciência e Tecnologia de Genômica para o Melhoramento de Citros (INCT-Citros; National Council for Scientific and Technological Development, CNPq 465440/2014-2 and FAPESP grant #14/50880-0).

**Acknowledgments:** The authors are grateful to citrus grower Azevedo for their yield and support. We also thank Grupo de Pesquisa em Citricultura (GD Citrus) at IAC for its support in the yield, and Laboratório de Microbiologia do Solo da Universidade de São Paulo (ESALQ) for technical support. A.C.C.A. acknowledge the Coordenação de Aperfeiçoamento de Pessoal de Nível Superior (CAPES, Brazil, #1794851). D.M.-J., F.A.d.A., and R.M.B. acknowledge the Conselho Nacional de Desenvolvimento Científico e Tecnológico (CNPq, Brazil) for fellowships granted. S.C. acknowledges FAPESP (#2018/24049-2) and CAPES (Finance Code 001).

**Conflicts of Interest:** The authors declare no conflict of interest.

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
