# Peer review of "Implication of Urochloa spp. Intercropping and Conservation Agriculture on Soil Microbiological Quality and Yield of Tahiti Acid Lime in Long Term Orchard Experiment"

_agriculture, doi:10.3390/agriculture10110491_

Round 1

Reviewer 1 Report

Revision of the manuscript entitled “Implications on the soil microbiological quality and  Tahiti acid lime orchard with different mower type in Urochloa spp. intercropped” by Dr. Ana Carolina Costa Arantes and coworkers (reference; agriculture-945703-peer-review-v1).

In this work the authors examine the long term effect of three treatments (two Urochloa spp.; conventional vs. ecological biomass cover; and application or not of herbicide glyphosate) on soil microbial populations/activity and on the yield of citrus lime trees.

This work is interesting for several reasons: first of all, it is a long term study; secondly, it involves trees; and in the third place, is a field experiment in an orchard. In this particular, not so many studies are performed on trees and for so long periods.

However, in my opinion, the main failure of this work is the absence of appropriate controls, i.e., a treatment without intercropping, without any type of biomass addition and without glyphosate. It would have been necessary in order to know the real effect of the treatments with regard to the yield and microbial populations/activities is non-treated soils with the same lime trees.

However, not forgetting this last comment, still some conclusions can be driven from the study. Nevertheless, major revision is still required in my opinion.

In the first place, the objectives must be clearly settled at the end of the introduction. It seems clear that other people had already demonstrated that biomass application increased lime yield in the system Urochloa-lime, and that conservative (ecological) mower ameliorates soil quality and microbial activity (referred in the introduction). So the objective is not comparing with the No-treatment, but comparing different treatments (two species; different types of tillage/mowers; and herbicides).

In this particular, an alternative and maybe more clear title could be “Implication of Urochloa spp. intercropping and biomass and herbicide application on soil microbiological quality and yield of Tahiti acid lime in long term orchard experiment” or “Implication of Urochloa spp. intercropping and ecological management (or conservation agriculture) on soil microbiological quality and yield of Tahiti acid lime in long term orchard experiment”. Of course these are just suggestions and must not be obligatorily considered by authors.

Once the objectives are stated, the results are correctly presented and interpreted. However, we can consider one aspect.  In field experiments, and due to the difficulty of controlling many variables such as rain, temperature, etc. sometimes using a probability of p< 0.1 is allowed (Gerhardt et al., 2009; Dary et al., 2010). If you try to do the statistical study in these conditions, maybe more differences can be observed and the conclusions raise more clearly.

K.E. Gerhardt, X.D. Huang, B.R. Glick, B.M. Greenberg, Phytoremediation and rhizoremediation of organic soil contaminants: potential and challenges, Plant Sci. 176 (2009) 20–30.

Dary, Mohammed & Chamber, Manuel & Palomares, A.J. & Pajuelo, Eloisa. (2010). “In situ” phytostabilisation of heavy metal polluted soils using Lupinus luteus inoculated with metal resistant plant-growth promoting rhizobacteria. Journal of Hazardous Materials. 177. 323-30. 10.1016/j.jhazmat.2009.12.035.

Other questions regarding the results are:

The first question arises from results of intercropping. From the third-fourth year it is clear that the production of biomass decreased, particularly in the conventional management. Could it be more interesting changing the intercrop after 3-4 years? What other(s) intercrop(s) could be used with lime?

Bacterial and fungal populations:

In Figure  4, how the abundance of bacteria/fungi was estimated? how it is calculated the log of ITS copy number?  How the diversity? It would have been much more interesting to show some RFLP profiles (even as supplementary information) in order to give an idea of the bacterial and fungal diversity. In Figure 4, X axis soil instead for solo

On the other hand, it is clear that the work is done for long period but still RFLP seems old technique (it does not mean that it is not valid to assess diversity), but massive sequencing could have provided much more information on the bacterial and fungal taxa. This aspect maybe must be discussed, the necessity of further analyses of microbial taxa (genomes of many of the resulting taxa are probably known, allowing toe stablish correlations with N2 fixers or P-solubilizers).

There is little effect of variables on bacterial or fungal abundance. The lack of the control without the variables (no intercropping, no tillage, no glyphosate) makes difficult the interpretation of results, since only little differences are observed. Even though, some differences are observed, statistically significant.

Regarding the abundance of diazotrophs and P solubilizers (putatively benefiting tree growth and/or productivity) only minor changes in log of copy number are observed (little effect of the intercropping species and the biomass application method). This could be a consequence of these results belonging to year 9, when the biomass of intercrops was severely diminished. Maybe using a probability of p<0.1 could be adequate. Also the aspect of PCA correlations maybe change upon considering p<0.1.

Data seem to depend more on the particular year (maybe climatologic conditions, etc.) so the 7th year seems to be particularly good, but a trend is difficult to be observed. However, the yield was not higher at year 7th, but later on, in years 8-9. Do you have some explanation for that?

It seems also that the effect of glyphosate persists in the soil for some years since no differences between the treatments are observed for the first 6 years. However, at longer periods some differences are observed. Maybe the degradation is slow.

Other comments

Line 151: Na2CO3

Author Response

Dear Reviewer 1, we appreciate the opportunity to review and your time spent on our manuscript. All your contributions and questions were important for the improvement of the article, we understand that they must be carried out and we are grateful to understand the importance of the work for the time of study, the use of trees and the experimental field. Corrections were made in red in the text, and the questions are answered.

Reviewer 2 Report

 Implications on the soil microbiological quality and Tahiti acid lime orchard with different mower type in Urochloa spp. intercropped

Lines

Original text

Comment/suggestion

General comments

The article is of great interest to the readers of Agriculture, combining many soil microbiological characteristics and providing plentiful data, useful for many speciealists. So it is publishable, but only after some editing, both terms- and language-wise.

Major comments

2-4, Title

Implications on the soil microbiological quality and Tahiti acid lime orchard with different mower type in Urochloa spp. intercropped

The title definitely needs editing. I suggest something along the lines like “Microbiological assessment of soil quality under a Tahiti acid lime orchard with different Urochloa intercrops and mowing”

102-107

Prior to giving soil chemical properties, the authors should provide soil name according to 1) national and b) international (IUSS, 2015) classifications.

599-604

Conclusion as a genre, alongside with the most important result of the study, should sketch its implications for a broader field and indicate avenues for further research.

The entire text

Although the format of electronic publication with MDPI has no restrictions on the size of the manuscript, the general rule of brevity being an asset still stands; so I would recommend to cut on holding forth in those places, where the authors find plausible. Otherwise the attention of readers somewhat dissolves.

Minor comments

24

Urochloa

I believe it is necessary to supply the common name of the plant as well

34

microbial soil

Well, rather interesting term. I encounter it for the first time and would like the authors to define it.

39

In the agriculture

In agriculture

40

 to the plants and to facilitate

to plants, thus facilitating

41

for measuring

for assessing (the are no methods for measuring soil quality, only for assessing it using a range of variables)

56

anthropic

This seems to be another unconventional term: can the authors please specify it?

88-89

Soil microbiological indicators can be used to assess their quality,

What does “their” refer to? It is unclear from the text; please, rewrite.

147

diazotrophocs

What are those? Please, correct.

Figure 2

Why does Urochloa biomass deposition vary greatly with orchard age?

Figure 7

Why did a sharp fruit yield occur in 8 yo orchard?

Technical comments

20

as well

including

21-22

effects in citrus was

the effects of such practices in soil under citrus orchards were

22-23

the soil microbiological attributes and effects on

effects of mowing and intercrop species on soil microbiological characteristics under a  

23

The orchard were planted in minimum tillage, intercropped two

The orchard was planted using minimum tillage and intercropped with two

24

Urochloa spp.

Urochloa

24

two mowers of Urochloa spp. biomass

two types of mowers for Urochloa biomass

25

use the herbicide in citrus rows, with assessments for up to 10 years.

herbicide application. The study was conducted for 10 years.

26

 Ecological promoted highest deposition of Urochloa spp. biomass that provided lowest disturbance to the soil microbial activity, improving, on average of the all years,

The ecological mower made the bigger deposition of the intercrop biomass, thus providing the lowest disturbance of soil microbial activity and increasing, on average of all years, the

28

the gene 16S

16S rRNA genes

29

and provided above 20%

and providing ca. 20%

30

 with ecological

in combination with the ecological mowing

31

 There was little influenced by the use of the herbicide.

There was little influence of the herbicide.

31-33

Then, the use of intercropped U. ruziziensis in citrus orchard, managed with the ecological mower is a sustainable practice of the conservation agriculture for soil microbiological quality and citrus production.

We conclude that the use of U. ruziziensis ias an intercrop in citrus orchards, subjected to the ecological mowing can be recommended as for improving and sustaining soil quality and citrus fruits production.

47-48

The microbial biomass, the labile fraction of the soil, is considered the active state of the organic matter in the soil

Soil microbial biomass is the labile fraction of soil organic matter,

49, 61, 65

the soil

soil

52

the increase in the yield

the increased yields

52-53

The 52 microbiological activity and the composition of the microbial community

Microbiological activity and microbial community composition

57

to the soil

to soil

57

an environment more balanced and sustainable

a more balanced and sustainable environment

58

to soil microorganisms

in soil microbial variables

60

in the citrus row

in citrus rows

60

management

practice

62

route, can cause

route can cause

69

Urochloa, such as in

Urochloa, as in

71

with intercropped

with intercrops

73

the maintenance of crop

stable crop

76

(19.6 million tons)

with 19.6 million tons

91

carried out

studied

92

is to

was to

92

the effect of a intercropped

the long-term effect of Urochloa intercropping

93-94

plants, with four different management strategies of Urochloa spp., over a long  period of time

plants

94-95

the effects on fruit yield Tahiti acid  lime

Tahiti acid lime fruit yield

109

sowing

sown

114

treatment and the

Treatment. The

108, 117

Poaceae

Poaceae (check, please, for consistency, as everywhere throughout the text should be the same)

134

afterwards, the biomass

afterwards  the biomass

138

yield

the yield

Figure 3, 4,5 axes titles, line 283 (elsewhere)

g solo-1

g-1 soil

Author Response

Dear Reviewer 2, we are very excited to have had the opportunity, given by the editor, to review our manuscript. We carefully considered your comments, and again, we want to extend our appreciation for taking the time and effort necessary to provide such insightful guidance. We have addressed all the indicated issues in the reviewed version.

Reviewer 3 Report

In this study, the effect of intercropping, tillage and herbicide use on soil quality and microbial activity was analyzed over the course of 10 years and correlated with the effects on the yield of Tahiti acid lime orchard cultures. I have several comments, which are mainly focused on improving the readability and understandability of the manuscript.

Major comments:

  • English style, grammar should be significantly improved. The authors also need to pay attention to spelling mistakes.
  • In some sections of the manuscript, the authors need to be more cautious with the conclusions drawn from the results provided in the manuscript. For example, in Line 280, “The Urochloa species did not influence microbial biomass carbon (MBC)”- the authors cannot verify this statement since no negative control (i.e. no Urochloa sown) was included in their assays.
  • The discussion is excessively long and unfocused. In its current state, it is difficult to contextualize the results obtained and to draw conclusions from them. The authors should make an effort to write a more focused discussion in order to better provide the reader with a general view of the contribution of their study to the field.

Specific comments

  • Include sequence of oligonucleotides used in this study (e.g. 8 fm, 926r, ITS1F, ITS4, etc.) at least as supplementary material.
  • All characters in the genes names must be italicized (e.g. nifH, phoD, etc.)

Author Response

Dear Reviewer 3, we are very grateful and thank you for your time and reading in our manuscript. All your questions and corrections were accepted and corrected throughout the text, and marked in green. Your questions are answered.  

Round 2

Reviewer 1 Report

I accepted the revision of Manuscript ID: agriculture-945703